# Molecular Mechanisms of Neurogenic Lower Urinary Tract Dysfunction after Spinal Cord Injury

**DOI:** 10.3390/ijms24097885

**Published:** 2023-04-26

**Authors:** Nobutaka Shimizu, Tetsuichi Saito, Naoki Wada, Mamoru Hashimoto, Takahiro Shimizu, Joonbeom Kwon, Kang Jun Cho, Motoaki Saito, Sergei Karnup, William C. de Groat, Naoki Yoshimura

**Affiliations:** 1Department of Urology, University of Pittsburgh School of Medicine, Pittsburgh, PA 15213, USA; nshimizu@kochi-u.ac.jp (N.S.);; 2Pelvic Floor Center, Kochi Medical School, Kochi University, Nankoku 783-8505, Japan; 3Department of Pharmacology, Kochi Medical School, Kochi University, Nankoku 783-8505, Japan; 4Department of Pharmacology and Chemical Biology, University of Pittsburgh, Pittsburgh, PA 15213, USA

**Keywords:** spinal-cord injury, detrusor overactivity, detrusor–sphincter dyssynergia, fibrosis, C-fiber afferent, Aδ-fiber afferent

## Abstract

This article provides a synopsis of current progress made in fundamental studies of lower urinary tract dysfunction (LUTD) after spinal cord injury (SCI) above the sacral level. Animal models of SCI allowed us to examine the effects of SCI on the micturition control and the underlying neurophysiological processes of SCI-induced LUTD. Urine storage and elimination are the two primary functions of the LUT, which are governed by complicated regulatory mechanisms in the central and peripheral nervous systems. These neural systems control the action of two functional units in the LUT: the urinary bladder and an outlet consisting of the bladder neck, urethral sphincters, and pelvic-floor striated muscles. During the storage phase, the outlet is closed, and the bladder is inactive to maintain a low intravenous pressure and continence. In contrast, during the voiding phase, the outlet relaxes, and the bladder contracts to facilitate adequate urine flow and bladder emptying. SCI disrupts the normal reflex circuits that regulate co-ordinated bladder and urethral sphincter function, leading to involuntary and inefficient voiding. Following SCI, a spinal micturition reflex pathway develops to induce an overactive bladder condition following the initial areflexic phase. In addition, without proper bladder–urethral-sphincter coordination after SCI, the bladder is not emptied as effectively as in the normal condition. Previous studies using animal models of SCI have shown that hyperexcitability of C-fiber bladder afferent pathways is a fundamental pathophysiological mechanism, inducing neurogenic LUTD, especially detrusor overactivity during the storage phase. SCI also induces neurogenic LUTD during the voiding phase, known as detrusor sphincter dyssynergia, likely due to hyperexcitability of Aδ-fiber bladder afferent pathways rather than C-fiber afferents. The molecular mechanisms underlying SCI-induced LUTD are multifactorial; previous studies have identified significant changes in the expression of various molecules in the peripheral organs and afferent nerves projecting to the spinal cord, including growth factors, ion channels, receptors and neurotransmitters. These findings in animal models of SCI and neurogenic LUTD should increase our understanding of pathophysiological mechanisms of LUTD after SCI for the future development of novel therapies for SCI patients with LUTD.

## 1. Introduction

Peripheral nerve functions during urinary storage and voiding are integrated by complex neural pathways in the cerebrum, brainstem, and spinal cord. For instance, during urinary storage, sensory inputs from the bladder via Aδ-fiber afferents carried through the pelvic nerve excites sympathetic nerves in the thoracolumbar spinal cord, causing contractions of bladder-neck and urethral smooth muscles, and relaxation of bladder smooth muscles. At the same time, excitation of the sacral Onuf’s nucleus also causes contraction of the external urethral sphincter via somatic axons carried through the pudendal nerve [1,2]. In contrast, during voiding, afferent signals from the bladder carried in the periphery via Aδ-fiber afferents ascend the spinal cord, and excite the relaying micturition centers such as the periaqueductal gray (PAG) and the pontine micturition center (PMC), from which descending fibers cause excitation of the parasympathetic nerves and inhibition of the sympathetic and somatic nerves, leading to efficient voiding without post-void residual urine.

Spinal-cord injury (SCI) at the level rostral to the lumbosacral spinal cord impairs voluntary and supraspinal control of micturition, initially resulting in an areflexic bladder and urine retention. Then, at the later phase, neurogenic detrusor overactivity (DO) and lack of detrusor and urethral sphincter coordination (termed detrusor sphincter dyssynergia or DSD) occur, leading to incomplete voiding, bladder enlargement, and elevated intravesical pressure. The recovery of reflex bladder activity following SCI depends on the rearrangement of reflex pathways in the spinal cord, including modifications of the properties of bladder afferent pathways [3,4,5]. After SCI, LUT dysfunction (LUTD) during the storage phase (DO) and voiding phase (DSD and inefficient voiding) is the major focus for the treatment of urological problems in SCI patients (Figure 1). The primary goals of therapy and treatment for LUTD have been thoroughly discussed, including protection of the kidneys from progressive damage, maintenance of renal function, and reduction in urinary incontinence to improve the patient’s quality of life. Each patient’s treatment plan should be individualized based on the results of urodynamic examinations, considering their level of disability, physical and mental health, and urinary-tract function [6]. SCI causes direct damage to axons, neuronal cell bodies, and glia, resulting in a loss of function below the injury site. In addition, the lesion induces an inflammatory response that contributes to secondary tissue damage, resulting in additional functional loss. Therefore, reducing inflammatory responses after SCI is a worthy therapeutic objective [7,8]. Using SCI animal models, previous studies, including ours, have sought to elucidate the pathophysiological changes in neural pathways or end-organ function and identify new therapeutic targets of LUTD following SCI. This article reviews the recent basic research findings regarding the molecular mechanisms underlying SCI-induced LUTD, focusing on normal roles and SCI-induced alterations in the functions of various molecules such as growth factors, ion channels, receptors and neurotransmitters.

## 2. Neurotrophic Factors

### 2.1. NGF (Nerve Growth Factor)

SCI can elicit morphological alterations in bladder afferent pathways, particularly in the C-fiber cell population, as well as an increase in the expression of CGRP and TRPV1 in bladder afferent neurons in rodents [9]. Two–three weeks after SCI, the density of CGRP-positive fibers in lamina 1 and 2 gradually increased to reach a level of around 50% over the original density at four weeks after SCI [10]. Neurotrophic factors such as NGF, BDNF, NT-3, and NT-4 are essential for various types of neuronal plasticity, including the emergence of LUTD in SCI [11,12,13,14]. Among them, NGF is a well-studied neurotrophic factor that controls the development and maintenance of the nervous system. Following SCI, DSD-induced overdistention of the bladder promotes NGF synthesis in the bladder, which appears to increase the excitability of C-fiber bladder afferent pathways, resulting in neurogenic DO [13,15,16,17] because the capsaicin-induced desensitization of C-fiber afferents inhibits DO in SCI rats and mice without affecting voiding contractions [18,19]. Furthermore, neutralization of NGF suppresses DO in rats and mice with SCI, similar to the effect of C-fiber afferent desensitization with capsaicin or resiniferatoxin [15,17,19]. Due to axonal transport of increased NGF in the bladder through afferent pathways, NGF levels are increased in cell bodies and spinal projections of dorsal root ganglia (DRG) neurons [11,17,20]. In addition, NGF levels are increased in the bladder of rodents with a partially obstructed urethra and in the urine of humans with bladder outlet obstruction, suggesting that overdistention of the bladder may be responsible for NGF overexpression in the bladder. [21,22]. Furthermore, chronic administration of NGF into the spinal cord or the bladder wall of rats promotes bladder overactivity and increases the firing rate of capsaicin-sensitive bladder afferent neurons [13,20,23,24].

TrkA, which is a high-affinity receptor for NGF, is predominantly expressed in C-fiber afferent neurons, and its expression is increased in rat-bladder afferent neurons after SCI [25,26]. Anti-NGF antibody treatment in mice with SCI decreased C-fiber-dependent non-voiding contractions (NVCs) during bladder filling and hyperexcitability of capsaicin-sensitive C-fiber bladder afferent neurons [17,19,27].

Changes in ion channels have been implicated in the actions of NGF. It has been shown that treatment with anti-NGF antibodies which reduces the hyperexcitability of C-fiber bladder afferent neurons is caused at least in part by the restoration of A-type K^+^ channel activity, which is decreased after SCI in mice [27] and rats [28]. SCI can also induce NGF-dependent plasticity of voltage-gated Na^+^ channels, shifting from a TTX-resistant subtype to a TTX-sensitive subtype, leading to bladder afferent hyperexcitability, which is likely to be an underlying mechanism of LUTD after SCI [18]. Our recent study using SCI mice also demonstrated that this SCI-induced Na^+^ channel plasticity was associated with TTX-resistant Nav1.8 downregulation and TTX-sensitive Nav1.7 upregulation in bladder afferent neurons, which are also reversed by NGF neutralizing treatment. These results suggest that these Na^+^ channel subunits among nine different Na^+^ channel subunits (Nav 1.1–1.9) could be potential therapeutic targets for new treatments of SCI-induced LUTD [29] (Figure 2).

### 2.2. p38 MAP Kinase

Effects of NGF are known to be mediated by second messenger signaling pathways involving phosphorylation and stimulation of a serine-threonine kinase (p38 MAP kinase) [30], which mediates cellular responses to a variety of chemical and physical insults [31]. NGF-mediated activation of p38 MAPK increases the expression of the TRPV1 receptor in DRG neurons [30]. Phosphorylated p38 MAPK is expressed primarily in small-to-medium-sized DRG neurons, which may correspond to C-fiber and some Aδ-fiber afferent neurons that innervate the bladder [32]. Our study using a novel herpes simplex virus vector-mediated neuronal labeling technique indicated that SCI in mice increases the TRPV1-expressing C-fiber afferent population and decreases the average cell size of TRPV1-expressing cells after SCI [9]. These results suggest that SCI induces de-novo expression of TRPV1 in small-sized C-fiber bladder afferent neurons, thereby increasing afferent excitability. Thus, NGF-induced phosphorylation of p38MPAK may contribute to TRPV1 overexpression in small-sized, C-fiber bladder afferent neurons after SCI. In SCI mice, inhibition of p38 MAPK reduced DO evident as reduced NVCs during the storage phase and improved voiding evidenced by increased voided volume and micturition pressure during the voiding phase, although it did not affect DSD during voiding bladder contractions. Thus, p38 MAPK signaling pathways may play an important role, at least in part, as a downstream signaling mechanism activated by NGF, in the emergence of LUTD after SCI [33].

### 2.3. BDNF (Brain-Derived Neurotrophic Factor)

BDNF levels in the rats’ and mice’s bladder and spinal cord were increased at 4–5 days following SCI [34,35,36]. By acting primarily on TrkB, a high-affinity, ligand-specific receptor of BDNF found on numerous neurons in the spinal cord and primary afferent pathways, BDNF promotes various physiological and pathological changes [37]. Our study using a novel herpes simplex virus vector-mediated neuronal labeling technique indicated that in SCI in mice, the number of NF200 (A-fiber marker) vector-labeled neurons was decreased in L6 DRG after SCI [9]. Although TrkB is expressed in both large (Aδ-fiber) and small (C-fiber) primary afferent neurons, after SCI, only large bladder afferent neurons exhibited increased TrkB expression [38,39]. A previous study using SCI rats showed that BDNF sequestration improved bladder function at later stages, suggesting that BDNF contributes DO maintenance in chronic SCI; however, inhibiting BDNF during the post-SCI spinal-shock period increased bladder overactivity, indicating that BDNF inhibits the emergence of C-fiber-mediated bladder overactivity in the early phase of SCI [40].

We have also reported that neutralization of BDNF using anti-BDNF antibodies in 4-week SCI mice improved voiding efficiency with increased voided volume [35,36]. In these studies, after BDNF neutralization, periodic decreases in the electromyography (EMG) activity of the external urethral sphincter (EUS) during voiding bladder contractions were enhanced to increase notch-like reductions in intravesical pressure, which corresponded with fluid elimination from the bladder, in SCI mice [41]. Thus, BDNF neutralization is likely to enhance urethral synergic relaxation during the voiding phase in SCI mice [35] (Figure 2), suggesting that anti-BDNF therapy is effective for the treatment of SCI-induced DSD whereas anti-NGF therapy is useful for the treatment of DO in the storage phase.

### 2.4. p75 Neurotrophin Receptor

Before their release, mature BDNF and NGF are processed enzymatically within secretory vesicles from their pro-neurotrophin precursors (proBDNF/proNGF) [42]. Under pathological conditions, their overexpression exceeds the processing rate, resulting in the uncontrolled release of pro-neurotrophins, which vary from mature forms in their affinity for the p75 low-affinity neurotrophin receptor (p75^NTR^), whose expression is altered in the bladder and neural networks controlling micturition following SCI [43]. Moreover, proBDNF/proNGF activates the p75^NTR^-sortilin complex preferentially, and sortilin is a transport protein involved in intracellular cargo arrangement across membrane compartments. When activated by proneurotrophins, p75^NTR^ may cause apoptosis in various cell types. Previous studies reported that a modulator of p75NTR, LM11A-31, which can promote p75NTR-associated signaling through disinhibition and/or activation of the TrkA/BNTR-p75 complex, ameliorates DSD and DO in SCI mice and also blocks SCI-related urothelial damage and bladder-wall remodeling [44,45,46]. Thus, drugs targeting p75^NTR^ could have therapeutic potential for the treatment of SCI-induced LUTD such as DSD and DO.

## 3. Transient Receptor Potential (TRP) Channels

### 3.1. TRPV1 and TRPA1 Receptors

TRPV1 or TRPA1 receptors in the suburothelial nerve fibers are implicated in C-fiber bladder hyperexcitability, contributing to neurogenic DO in SCI [47,48]. After SCI, C-fiber bladder afferents in cats, which normally do not respond to bladder distention, become mechanosensitive and induce involuntary micturition [43]. In cats with chronic SCI, desensitization of TRPV1-expressing C-fiber afferent pathways by subcutaneous administration of capsaicin, a C-fiber neurotoxin that binds TRPV1 receptors, can significantly inhibit DO during the storage phase. In contrast, this effect is not observed in reflex bladder contractions in intact spinal cats [43]. In SCI rats, both Aδ and C-fiber afferents can elicit bladder reflexes [36], and increased excitability of the latter induces DO because desensitization of C-fiber afferents by systemic administration of capsaicin significantly suppressed NVCs during the storage phase without affecting the voiding reflex [43]. These results indicate that C-fiber afferents contribute to neurogenic DO generation during storage, whereas A-fiber afferents are involved in urination in rats with SCI. Furthermore, a novel HSV vector-mediated neuronal labeling technique revealed that SCI induces expansion of the TRPV1-expressing C-fiber cell population, possibly contributing to C-fiber afferent hyperexcitability [9]. TRPV1 upregulation is closely related to NGF overexpression after SCI. In addition, the systemic administration of TRPV1 or TRPA1 antagonists reduced both bladder contraction frequency and bladder overactivity in SCI animals [47,48]. Neutralization of NGF decreases NGF levels in the bladder and the spinal cord, inhibits bladder overactivity, and reduces TRPV1 and TRPA1 expression in L6-S1 DRG [17,49].

Thus, it is likely that TRPV1 or TRPA1 receptor-targeting therapies could be effective for the treatment of SCI-induced bladder overactivity. Herpes simplex virus (HSV) has a natural property that it is transported to afferent pathways from primary infection sites, which could offer an organ-specific treatment for sensory nerve-related diseases. A previous study also indicated that deletion of the channel pore-forming domain of the TRPV1 receptor inhibits the assembly including an aqueous pore, which blocked the channel function [50]. We previously examined the effect of HSV vector-mediated gene delivery of non-functional, poreless TRPV1 (PP1α) in storage and voiding dysfunction in SCI mice [51]. HSV vectors injected into the bladder wall are transported to L6 DRG neurons through bladder afferent pathways, and HSV vector-mediated gene delivery of PP1α significantly reduced DO after SCI as well as the phosphorylated level of TRPV1 in the bladder. Gene therapy with replication-deficient HSV vectors encoding poreless TRPV1, which can suppress TRPV1 receptor activation in the bladder and bladder afferent pathways, could be a potential treatment that can avoid systemic adverse events for neurogenic LUTD/DO after SCI [51].

### 3.2. TRPC Channels (TRPC1, TRPC3, and TRPC6)

In contrast to TRPV1 channels, the expression and regulation of TRPC channels in micturition-related afferent pathways are poorly understood. TRPC1, TRPC3, and TRPC6 are most abundantly expressed in the cluster of adult-mouse DRG neurons that transmit sensory information to the spinal cord [52]. The expression levels of TRPC3 and TRPC6 in L6-S1 DRG of SCI mice were lower than in spinal intact mice [49]. TRPC1 was expressed in the neurofilament 200-positive large-sized subclass of DRG neurons, while TRPC3 mRNA expression, which stained up to 35% of DRG neurons, was almost only found in non-peptidergic, isolectin B4 (IB4)-positive small-sized neurons that were mostly TRPV1-negative [49]. As stated previously, the expansion of TRPV1-positive bladder afferent neurons following SCI occurred in small-sized neurons in our previous HSV vector-tracing study [9]. Thus, a reduction in TRPC3/C6 and increased TRPV1 may be induced in small, non-peptidergic C-fiber bladder afferent neurons following SCI.

Our study showed that anti-NGF antibody treatment normalized the expression of TRPV1, TRPC3, and TRPC6 but not TRPC1 in L6-S1 DRG of SCI mice; therefore, NGF may play a lesser role in modulating TRPC1 expression following SCI [49]. In addition, TRPC3/6 may play an inhibitory role in regulating cell excitability of non-peptidergic C-fiber bladder afferent neurons. However, the functional role of TRPC channels in the control of micturition is still unclear; thus, their reduction could contribute to C-fiber hyperexcitability after SCI, though future studies are needed to support this point.

## 4. Mechanosensitive Channels

### 4.1. ASICs (Acid-Sensing Ion Channels)

In bladder afferent pathways, acid-sensing ion channels (ASICs) expressed by sensory neurons may also be a molecular target of BDNF. Although ASICs were first identified as receptors triggered by a decrease in extracellular pH [53,54], it was subsequently shown that they also have mechanosensory activities [55]. In DRG, ASIC3 receptors are expressed on TRPV1-expressing, unmyelinated C-fiber afferent neurons and also on mechanosensitive, myelinated A-fiber afferent neurons [56,57]. The ASIC2 channel is a target for BDNF signaling and regulates sensory mechanotransduction [54]. In our recent study, ASIC2 and ASIC3 transcripts in L6-S1 DRG were upregulated in SCI mice and were suppressed after BDNF neutralization which improved SCI-induced DSD and inefficient voiding. On the other hand, TRPV1 expression in L6-S1 DRG, which was increased following SCI, was unaffected by BDNF neutralization. [35]. Thus, it is assumed that BDNF enhances mechanotransduction through ASIC channels expressed in Aδ-fiber bladder afferent pathways to facilitate the bladder-to-EUS reflex, leading to DSD and inefficient voiding after SCI.

### 4.2. Piezo Channels

Piezo1 and Piezo2 ion channel proteins are predominantly expressed in the bladder and neural pathways, respectively, in the lower urinary tract. Piezo2 is mechanically activated and expressed in a subpopulation of low-threshold mechanosensory neurons that detect stimulus movement direction [58,59]. A decrease in BDNF expression is associated with morphological polarization of Aδ-fiber low-threshold mechanoreceptive neurons, leading to the failure of direction-selective responses in these neurons [60,61]. Mice-bladder epithelium and afferent pathways express Piezo1 and Piezo2 channels, which can regulate low-threshold bladder-stretch sensing and micturition reflexes [62,63]. Analyses of the time-course of receptor changes after SCI in mice revealed that TRPV1 and ASIC1-3 in L6-S1 DRG increased early during the 2–4 weeks postinjury, but Piezo2 in L6-S1 DRG increased in the later phase at six weeks postinjury [64]. In our preliminary experiments, in mice with SCI, intrathecal administration of GxMTx4 (a Piezo blocker) enhances voiding efficacy and the duration of EUS-EMG reduction time during voiding bladder contractions. Thus, treatments suppressing mechanosensitive channels such as Piezo2 and ASICs that are expressed in Aδ-bladder afferent pathways might be effective for reducing voiding dysfunction due to DSD after SCI.

## 5. Neurotransmitters and Their Receptors

### 5.1. Purines: ATP and Inosine

SCI can produce changes in the bladder urothelium and smooth muscles, leading to stimulation of bladder afferents and induction of DO. This may occur in part via a peripheral action of ATP because ATP released from the urothelium can activate bladder afferents through P2X2/3 ATP receptors in a rat model of SCI [65,66]. A central action of ATP has also been demonstrated using a partial SCI rat model induced by bilateral dorsal lesions of the thoracic spinal cord. In this model, P2X7 ATP receptors were expressed at the spinal-cord injury site in CD11b-positive microglia cells and treatment with a P2X7 ATP receptor antagonist at the injury site reduced DO [67]. As the ATP release from the spinal cord during bladder distension is reportedly increased in SCI rats compared to spinal-intact rats [68], it is likely that the ATP-mediated excitatory mechanism in the spinal cord via P2X7 receptors expressed in microglia cells is enhanced to induce neurogenic DO after SCI.

Although other purines including adenosine have not been well-studied in the context of SCI-induced LUTD, recent studies by Adams and her colleagues demonstrated that systemic and intravesical application of inosine, which is generated extracellularly by deamination of adenosine or intracellularly through the action of 5′-nucleotidase on inosine monophosphate, improved DO in SCI rats [69,70]. In addition, this inosine-induced suppression of DO after SCI was associated with reduced spontaneous activity of bladder muscle strips via activation of adenosine A2B receptors [71]. Previous studies have reported that the enhanced intrinsic detrusor activity is linked to the increased firing of single-unit bladder afferent fibers after SCI [72,73]. Thus, these results suggest that inosine could be effective in treating complications of SCI such as DO.

### 5.2. Nitric Oxide (NO)

NO-cGMP pathways are known to be involved in the control of LUT function in multiple ways, including urethral and bladder-neck smooth-muscle relaxation, increased blood flow due to vascular smooth-muscle relaxation, and peripheral inhibition of afferent bladder activity. In addition, phosphodiesterase type 5 (PDE5) inhibitors such as tadalafil, which increase cellular levels of cGMP, have been used for the treatment of LUT symptoms in males with benign prostate hypertrophy.

Previous studies using SCI rodent models showed that expression levels of neuronal NO synthase (nNOS) is decreased in the urethra (mouse) [74] and increased in DRG afferent neurons and the spinal cord (rat) [75] suggesting the reduced and enhanced activity of the NO system in LUT efferent and afferent pathways, respectively. Due to administration of a soluble guanylate cyclase (sGC) activator, which directly increased the cGMP level in the urethra independent of NO, improved inefficient voiding along with the increase in EUS relaxations during voiding bladder contractions in SCI mice, it seems likely that activation of NO-cGMP pathways in the urethra is effective for the treatment of SCI-induced DSD [74]. Although it is not known whether sGC activation directly or indirectly induces synergistic relaxations of EUS striated muscles, the former might be the case as a previous study showed that PDE5 is expressed in rat EUS striated muscles more abundantly than in urethral smooth muscles [76]. In line with these basic research data, a previous clinical study reported that oral administration of NO donors significantly reduced EUS pressures at rest and during dyssynergic contraction in 15 SCI male patients, suggesting that NO might be effective for the treatment of DSD in SCI [77]. In addition, sGC activator treatment also improved DO evident as reduced NVCs during the storage phase along with the reductions in C-fiber afferent markers such as TRPV1 in L6-S1 DRG as well as ischemic and fibrosis-related markers in the bladder in SCI mice [74]. These results suggest that activation of NO-cGMP pathways in the bladder is effective for the treatment of C-fiber-dependent DO and bladder remodeling after SCI. Anti-ischemic and anti-fibrotic effects were also reported in SCI rats with PDE5 inhibitor (tadalafil) treatment [78].

In contrast to these peripheral, inhibitory effects of the NO system on LUT function, NO released from central afferent nerve terminals in the spinal cord seems to be excitatory to induce bladder overactivity because a previous study demonstrated that intrathecal NOS inhibitor administration reduced urinary incontinence by increasing bladder capacity in SCI rats [75].

Taken together, the NO-cGMP pathway, which is involved in the control of LUT function in separate ways at multiple levels, could be an effective target for the treatment of SCI-induced LUTD.

### 5.3. Inhibitory Neurotransmitters; GABA and Glycine

Glycine and gamma-aminobutyric acid (GABA) are the primary neurotransmitters inhibiting the micturition reflex at supraspinal and/or spinal sites through additive or synergistic inhibition of bladder activity [79]. It has been shown that SCI rats develop LUTD, such as DO or DSD, when glycinergic or GABAergic mechanisms in the lumbosacral spinal cord become hypofunctional [80,81,82]. Thus, glycine administered intrathecally, intravenously, or orally reportedly improves bladder and urethral dysfunctions in SCI rats [82,83]. In addition, intrathecal administration of muscimol and baclofen (GABA_A_ and GABA_B_ agonists, respectively) inhibits NVCs via inhibiting C-fiber bladder afferents and improves DSD in rats with SCI [80,81]. mRNA levels of glutamic acid decarboxylase (GAD), an enzyme implicated in GABA synthesis, are decreased in the spinal cord following SCI [81]. In addition, a time-dependent decrease was observed in the number of GABAergic neurons in the lumbosacral dorsal horn and lamina X after SCI [10], indicating that diminished GABAergic inhibition may contribute to developing neurogenic DO. Thus, gene delivery of GAD through nonreplicating HSV vectors inhibits DO and DSD in rats with SCI without affecting voiding contraction [84,85].

### 5.4. Serotonin (5-Hydroxytriptamine, 5-HT)

As previously reported, the serotonergic mechanisms at supraspinal or spinal sites are greatly involved in the inhibitory control of micturition [1]. Previous studies using SCI rat models showed that inefficient voiding and DSD evident as tonic EUS activity during voiding bladder contractions were improved by systemic or intrathecal activation of 5-HT receptors such as 5-HT_2A/2C_, 5-HT_7_ or 5-HT_1A_, and that the expression of 5-HT_2A/2C_ and 5-HT_7_ receptors were upregulated in lumbosacral cord motoneurons in the Onuf’s nucleus [86,87,88]. Thus, certain 5-HT receptor subtypes could be potential targets for the treatment of SCI-induced LUTD such as DSD.

### 5.5. Dopamine

Dopamine is known to be an important transmitter controlling various body functions including lower urinary tract at the supraspinal site [1]. However, recent studies by Hou and his colleagues using a rat model of SCI reported that spinal dopaminergic neurons are located in the autonomic nuclei and superficial dorsal horn of the L6-S3 spinal cord [89]. It has also been shown that this spinal dopaminergic machinery is involved in the control of the spinal micturition reflex following SCI through D1-like dopaminergic receptors that suppress tonic EUS activity to enable voiding and D2-like receptors that facilitate voiding [90,91]. Thus, modulation of the spinal dopaminergic system could be effective for the treatment of SCI-induced LUTD.

### 5.6. Neurokinins

In spinally intact rats, the destruction of lumbosacral spinal neurons expressing neurokinin-1 receptors (NK-1R) with an NK-1 ligand conjugated with saporin has no effect on the voiding reflex. However, it reduces the bladder irritant effects of intravesical capsaicin and decreases MVCs in SCI rats [16,92]. Similarly, intrathecal administration of a selective NK1R antagonist does not affect the micturition reflex in rats with intact spinal cords but inhibits it in rats with SCI [46]. These findings, along with the increased expression of substance P in the sacral parasympathetic nucleus region of SCI rats, suggest that NK-1R activation is implicated in the control of micturition in SCI animals [46].

In addition, NK-2R agonists act as bladder prokinetic agents and induce smooth-muscle contractions by stimulating NK-2R in smooth-muscle cells [93,94]. NK-2R agonists elicit dose-dependent bladder pressure increases with a quick onset and short duration [95,96,97]. Within minutes of administration, NK-2R agonists induce effective voiding in awake, spinal-intact dogs [97]. This effect is also observed in rats with acute SCI during anesthesia. SCI rodents with chronic administration of NK-2R agonists have consistent and efficient voiding [98]. Thus, NK receptors might provide therapeutic targets for treating SCI-induced LUTD.

### 5.7. β3-Adrenoceptors

The β3-adrenoceptor, located in the bladder smooth muscle, is involved in bladder relaxation [99]. In addition, it has been demonstrated that β-adrenoceptor activation of the urothelium causes nitric oxide release [100] and that CL316,243, a β3-adrenoceptor agonist [101], or nitric oxide [102,103] can inhibit bladder afferent nerve activity in spinal-cord-intact rats. In rats with SCI, β3-adrenoceptor agonists also reduce bladder afferent nerve firing, possibly due to the suppression of detrusor muscle micromotions [102,104].

Mirabegron is the first clinically available β3 agonist approved for use in adults with overactive bladder. Mirabegron was approved for medical use in Japan in 2011 and in the United States and in the European Union in 2012. Mirabegron has an affinity not only for β3-adrenoceptors, but also for β2-receptors to a lesser degree [105]. Mirabegron also exhibits α1-adrenoceptor antagonism and induces urethral relaxation in rodents [106]. In SCI rats, mirabegron inhibits NVCs, decreases maximum bladder contraction pressure, and reduces the residual volume [107].

In clinical practice, one randomized, placebo-controlled trial examined the mirabegron effect in patients with neurogenic LUTD suffering from SCI or multiple sclerosis. In this study, bladder volume at the first detrusor contraction and bladder compliance significantly improved after mirabegron treatment, whereas the increase in cystometric capacity and the decrease in maximal detrusor pressure were not statistically significant after treatment [108].

Vibegron is a new β3-adrenoceptor agonist approved for the treatment of overactive bladder in Japan in 2018 and in the United States in 2020. Treatment of SCI mice with vibegron reduced DO evident as the decreased number of NVCs and decreased mRNA levels in L6-S1 DRG for TRPV1, TRPA1, inducible nitric oxide synthase (iNOS) and activating transcription factor 3 (ATF3), an inflammatory cytokine, all of which were increased after SCI. These results indicate that β3 adrenoceptor activation by vibegron improved the SCI-induced storage dysfunction, possibly through the reduction in C-fiber-related receptor expression and inflammation-related markers in DRG [109]. In addition, vibegron delayed the first NVC and reduced collagen types 1 and 3, TGF-β1, and HIF-1α mRNA expression of the bladder in SCI mice [110], suggesting that vibegron treatment is effective not only for LUTD, but also tissue remodeling of the bladder after SCI.

We also investigated the efficacy of vibegron in 23 SCI patients and found that vibegron decreased maximal detrusor pressure and increased maximal cystometric capacity during the storage phase and also improved other CMG parameters, including bladder volume at first DO and bladder compliance [111]. One potential mechanism for vibegron-induced effects is that it can bind to muscarinic receptors, as shown in the rat bladder with a relatively higher affinity for the M2 subtype than M1 and M3 subtypes [112]. Taken together, these results suggest that vibegron might act through not only β3 adrenoreceptors, but also muscarinic receptors, in addition to direct or indirect suppression of C-fiber afferent activity to regulate bladder contractions. However, further studies are needed to understand the exact mechanisms through which vibegron improves SCI-induced DO.

## 6. Tissue Fibrosis

Bladder fibrosis is often related to urologic conditions, including chronic bladder inflammation, benign prostatic hyperplasia, and neurologic LUTD [113,114]. After SCI, DSD promotes bladder distention, followed by bladder ischemia and fibrosis [114]. In addition, the denervation of detrusor muscles after SCI might result in muscular atrophy, collagen deposition, and eventual fibrosis [115]. Furthermore, bladder fibrosis may worsen voiding dysfunction by decreasing detrusor contractility and a deposition of the extracellular matrix proteins might make the bladder wall thicker and less flexible. Changes in the organ structure may also hinder the interaction between neurons and detrusor smooth muscles [116].

The bladders of SCI rats exhibited increased transcript levels of type-3 collagen, hypoxia-inducing factor-1, transforming growth factor (TGF)-1, and fibroblast growth factor (FGF) [117,118]. Nintedanib, which suppresses the production of fibrosis-related factor receptors such as vascular endothelial growth factor (VEGF), fibroblast growth factor (FGF), and platelet-derived growth factor (PDGF), is approved as a treatment medication for patients with idiopathic pulmonary fibrosis. Systemic nintedanib therapy in SCI mice improved both storage and voiding dysfunctions, as shown by increases in voided volume per micturition and voiding efficiency [118]. Administration of nintedanib to SCI mice also reduced the expression of fibrosis-related markers in the bladder mucosa and detrusor, including TGF-1 and collagen types 1 and 3 [118]. After therapy with nintedanib, the elevated mRNA levels of TRPV1, TRPA1, P2X2, and P2X3 in the L6 and S1 DRG of SCI mice were considerably reduced, as was the collagen deposition (Masson’s trichrome staining) in the bladder [118]. Consequently, antifibrosis medication may ameliorate LUTD caused by bladder fibrosis following SCI. In addition, the nintedanib-induced alleviation of bladder storage dysfunction may be achieved by modulation of bladder C-fiber afferent activity, evidenced by the reduction in C-fiber afferent markers. Thus, these results suggest that anti-fibrosis therapy has potential as an effective treatment of SCI-induced bladder remodeling, thereby improving neurogenic LUTD in SCI.

## 7. Conclusions

Various studies have linked neurogenic LUTD after SCI in animals to plasticity in the peripheral and central nervous systems. Although hyperexcitability of C-fiber bladder afferents has been proposed as a major pathophysiological foundation of neurogenic LUTD, recent evidence suggests that plasticity of the Aδ-fiber bladder afferent pathways may be implicated at least in part in the pathophysiology of SCI-induced DSD and inefficient voiding. This review article identifies multiple variables leading to these neural plastic changes, which include neurotrophic factors, TRP and mechanosensitive channels, various neurotransmitter systems and tissue remodeling pathways, which might assist in selecting prospective therapy targets for both storage and voiding LUTD after SCI. Life activities would be significantly changed in patients with SCI, and the associated complications would also affect their physical and financial burdens. However, current medical technologies could not completely help them to return to their pre-injury healthy life [6]. More animal-based translational research is encouraged to develop innovative treatment approaches for humans with SCI-induced LUTD.

Summary: The hyperexcitability of C-fiber bladder afferents is an important pathophysiological cause of neurogenic lower urinary tract dysfunction in animal models of SCI, and several neuronal plasticities in the peripheral and central nervous systems are recognized. In addition, recent animal studies suggest that Aδ-fiber afferent pathways may be implicated in the pathophysiology of SCI-induced LUTDs and DSD. Identifying multiple variables leading to this neural plasticity might assist in selecting prospective therapy targets for LUTD after SCI (Figure 3). More animal-based translational research is required to develop new therapeutic modalities for the treatment of SCI-induced LUTD.

## Figures and Tables

**Figure 1 ijms-24-07885-f001:**
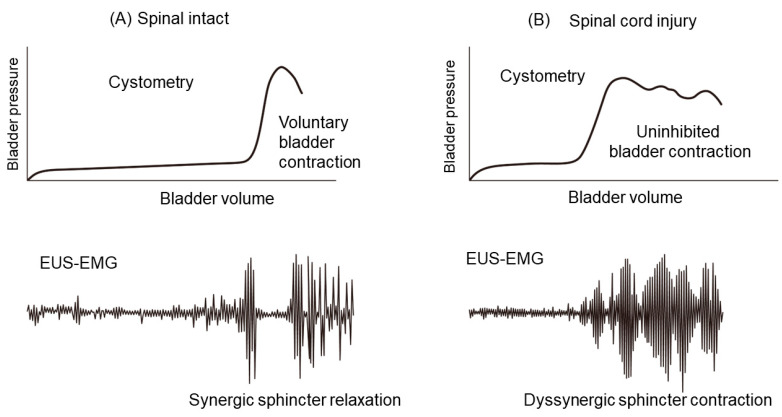
Human CMG and EUS-EMG recordings are representative. (**A**) Cystometry and EUS-EMG recordings under normal conditions reveal that voluntary bladder contraction induces synergic urethral relaxation. (**B**) After spinal-cord damage, cystometry and EUS-EMG recordings reveal uninhibited bladder contractions (detrusor overactivity) and detrusor sphincter dyssynergia. EUS-EMG: external urethral sphincter electromyography.

**Figure 2 ijms-24-07885-f002:**
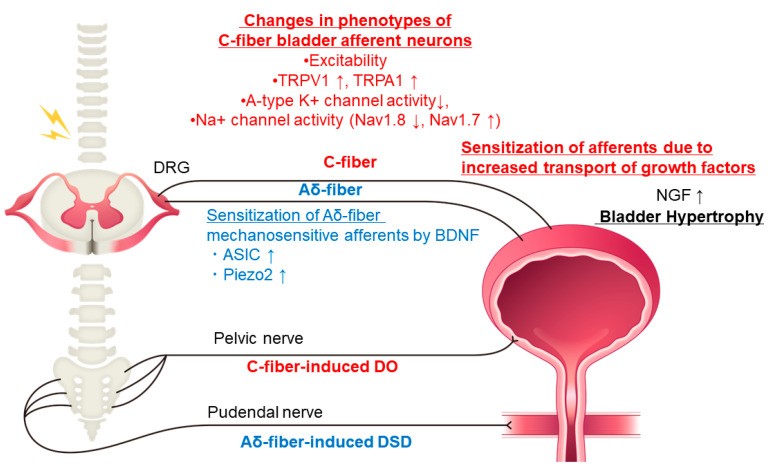
Diagram showing hypothetical mechanisms underlying storage and voiding dysfunction induced by increased expression of neurotrophic factors following SCI. Injury to the spinal cord causes DSD, leading to functional urethral obstruction, reduced voiding efficiency, urinary retention, and bladder hypertrophy, resulting in increased levels of NGF in the bladder. NGF is taken up by TrkA-expressing C-fiber afferent nerves and transported to lumbosacral DRG cells and central afferent nerve terminals. The levels of NGF are also increased in the spinal cord after SCI. NGF then sensitizes C-fiber bladder afferent pathways to cause or enhance neurogenic DO in SCI. BDNF is also increased in the bladder and the spinal cord after SCI. BDNF is expressed on larger sized bladder afferent neurons, presumably Aδ-fiber afferents, which express mechanosensitive receptors such as ASIC, and Piezo2. Hyperexcitability of Aδ-fiber bladder afferent pathways causes or enhances DSD, leading to inefficient voiding. Systemic application of BDNF antibodies reduces BDNF levels in bladder afferent pathways and improves DSD. SCI: spinal-cord injury, DSD: detrusor-sphincter dyssynergia, NGF: nerve growth factor, DRG: dorsal root ganglion, DO: detrusor overactivity, BDNF: brain-derived neurotrophic factor. Arrows indicate the increase or decrease of molecules.

**Figure 3 ijms-24-07885-f003:**
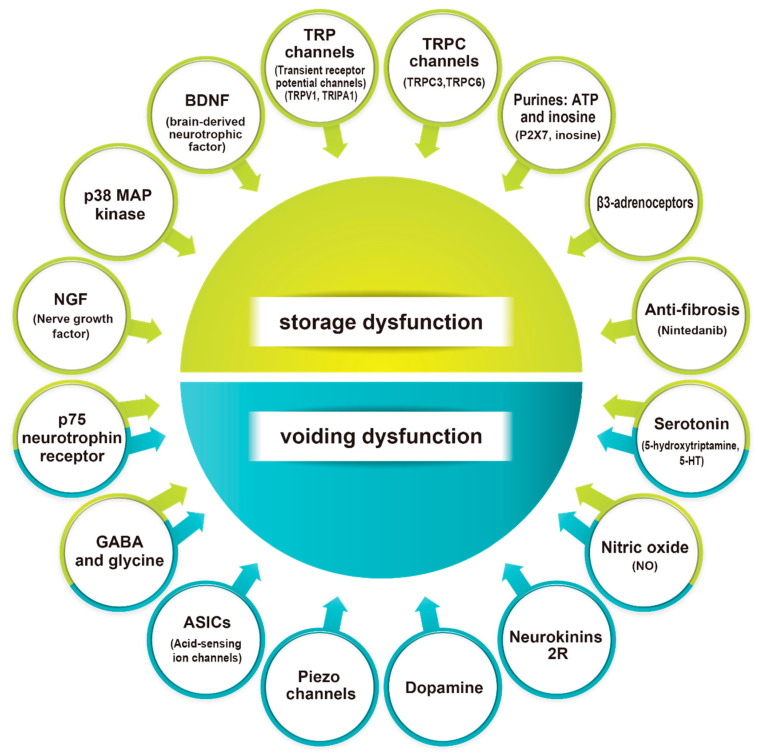
Summary of the potential molecular targets for the treatments of LUTD after SCI. Multiple targets for the better/new treatments of LUTD due to SCI were reviewed in this article. Some targets are responsible for the storage dysfunction, such as DO, and others are for the voiding dysfunction, such as DSD and inefficient voiding. These molecular targets could hopefully be translated for the development of future clinical treatment modalities of SCI patients with LUTD.

## Data Availability

Not applicable.

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
