# Peer review of "Molecular Mechanisms of Neurogenic Lower Urinary Tract Dysfunction after Spinal Cord Injury"

_ijms, 2023, doi:10.3390/ijms24097885_

Round 1
Reviewer 1 Report
It is a well written article that shed light on Molecular mechanisms of neurogenic lower urinary tract dysfunction after spinal cord injury. However, some points should be addressed and updated.
1. Some articles in the references should be updated, such as Ref 13, 27, 46, 53 and 55. The authors should avoid to cite articles in last century, especially for citing an old review article (Ref 46). For some classical concepts and theories, the authors could reference their recent work, such as Wada et al. Urol Sci. 2022;33(3):101-113.
2. To understand the knowledge of spinal cord injuries is to befit patients in clinics. Therefore, to address the current management of spinal cord injury patients is acquired to nourish this review in the aspect of translational research. Recently, Kuo has led a review article to discuss the management of neurogenic voiding dysfunction after spinal cord injury, which would be interesting to update the relationship between molecular mechanisms and urodynamic findings. (Chen YJ, et al. Urol Sci 2023, 34; 3-9.)
3. Several molecules and neurotransmitters discussed originated from the situation of neurogenic inflammation. The authors should introduce the concept of neurogenic inflammation in the section of introduction.
Author Response
Reviewer #1
Comments and Suggestions for Authors
It is a well written article that shed light on Molecular mechanisms of neurogenic lower urinary tract dysfunction after spinal cord injury. However, some points should be addressed and updated.
- Some articles in the references should be updated, such as Ref 13, 27, 46, 53 and 55. The authors should avoid to cite articles in last century, especially for citing an old review article (Ref 46). For some classical concepts and theories, the authors could reference their recent work, such as Wada et al. Urol Sci. 2022;33(3):101-113.
Response: We appreciate the reviewer’s comment and we do understand the concern about old references. According to the comment, we deleted Ref # 13, 27, 46, 53 & 55 as well as the last century articles.
- To understand the knowledge of spinal cord injuries is to befit patients in clinics. Therefore, to address the current management of spinal cord injury patients is acquired to nourish this review in the aspect of translational research. Recently, Kuo has led a review article to discuss the management of neurogenic voiding dysfunction after spinal cord injury, which would be interesting to update the relationship between molecular mechanisms and urodynamic findings. (Chen YJ, et al. Urol Sci 2023, 34; 3-9.)
Response: We have added the sentence as suggested (page 6, line 11-15 and page 29, line 15-18) with a reference (Chen YJ, et al. Urol Sci 2023, 34; 3-9).
- Several molecules and neurotransmitters discussed originated from the situation of neurogenic inflammation. The authors should introduce the concept of neurogenic inflammation in the section of introduction.
Response: We have added the sentence regarding inflammatory responses as suggested (page 6, line 12-17) with the following references (#7-8):
7.Anwar, M.A.; Al Shehabi, T.S.; Eid, A.H. Inflammogenesis of Secondary Spinal Cord Injury. Front Cell Neurosci 2016, 10, 98, DOI:10.3389/fncel.2016.00098.
8.David, S.; Zarruk, J.G.; Ghasemlou, N. Inflammatory pathways in spinal cord injury. Int Rev Neurobiol 2012, 106, 127-152, DOI:10.1016/b978-0-12-407178-0.00006
Reviewer 2 Report
Thank you for this important and well written report on, Molecular mechanisms of neurogenic lower urinary tract dysfunction after spinal cord injury. One suggestion, it would add, to have some spinal cord histological figures of c-fiber and a-delta fiber and receptor changes - before and after SCI - as published before.
Author Response
Reviewer #2
Comments and Sugg (estions for Authors
Thank you for this important and well written report on, Molecular mechanisms of neurogenic lower urinary tract dysfunction after spinal cord injury. One suggestion, it would add, to have some spinal cord histological figures of c-fiber and a-delta fiber and receptor changes - before and after SCI - as published before.
Response: We appreciate the reviewer’s comment and we added the sentence as suggested (page 7, line 7-11 and page 11, line 2-5) with the following reference (#10). We chose not to include histological figures as the readers can directly access those figures in the cited reference.
- Sartori, A.M.; Hofer, A.S.; Scheuber, M.I.; Rust, R.; Kessler, T.M.; Schwab, M.E. Slow development of bladder malfunction parallels spinal cord fiber sprouting and interneurons' loss after spinal cord transection. Experimental neurology 2022, 348, 113937, DOI:10.1016/j.expneurol.2021.113937.
Reviewer 3 Report
The authors present an extensive and well laid out review of molecular mechanisms in neurogenic bladder. They begin with discussion of neurotrophic factors, then TRPs, then mechanosensitive channels, then neurotransmitters, and finally a discussion on tissue fibrosis.
On our literature search this topic has been discussed but not reviewed/synthesized to the extent which the authors do so here therefore there is interest to the readership.
Some points of consideration for improvement: Could the authors explore specific genes that are implicated in neurogenic bladder after SCI and create a subsection on this? We would also like to see if there is any epigenetic control in this field. This can bring in the work of other researchers in this field.
Author Response
Editor’s comment
Thank you so much for having considered our journal as one platform for your
work. There are another *two issues* that need your assistance.
- Firstly, *Figure 3* was not mentioned in the main text, please add it.
Response: We have mentioned Figure 3 in the manuscript (page 30, line 10)
- Secondly, apart from making some revisions according to the reviewers'
comments, please also notice that there are some sections in your manuscript
that are highly matched to the article named "Current Knowledge and Novel
Frontiers in Lower Urinary Tract Dysfunction after Spinal Cord Injury: Basic
Research Perspectives" and article named "Neuroscience Letters", especially
the part marked the red number one (shown in *iThenticate report* in the
attachment), so you can modify your manuscript against the latter.
Response:
According to the comments, we rewrote the text that was overlapped with our previous publications (red text).
Round 2
Reviewer 3 Report
The authors have done a nice job addressing concerns raised by reviewers